# Multimodal Context-Aware Transformer with Visual Guidance for Automated 3D Annotation

## Abstract

The laborious nature of manual point cloud labeling drives the growing interest in 3D auto-annotation. The challenge is amplified by the sparse and irregular distribution of point clouds. This leads to the under-performance of current autolabelers, particularly with hard-to-detect samples (hard samples) characterized by truncation, occlusion, or distance. In response, we propose a multimodal context-aware transformer (MMCAT) as a 3D annotator using a small number of 3D annotations. MMCAT integrates 3D point cloud geometry with image-based semantic insights to improve 3D hard samples' annotations through 2D visual guidance. Our approach utilizes visual hints from three perspectives to integrate the 2D and 3D dimensions. Initially, we develop point and image encoders to align LiDAR and image data, establishing a unified semantic bridge between image visuals and point cloud geometry. Subsequently, our box encoder processes 2D box coordinates to improve accuracy in determining object positions and dimensions within 3D space. Finally, our multimodal encoders enhance feature interactions, improving point cloud interpretation and annotation accuracy, especially for challenging samples. MMCAT lies in its strategic use of 2D visual prompts to bolster 3D representation and annotation processes. We validate MMCAT's efficacy through extensive experiments on the widely recognized KITTI and Waymo Open datasets, particularly highlighting its superior performance with hard samples.

## 1 Introduction

3D point cloud technology, driven by LiDAR advancements, has advanced fields like autonomous driving and robotics, pushing forward 3D object detection. Groundbreaking 3D object detectors such as PointRCNN Shi et al. (2019), PointPillars Lang et al. (2019), and PAConv Xu et al. (2021) have significantly advanced the ability to identify objects within complex 3D environments. Yet, progress in this domain is hampered by the intensive labor and challenges to generate high-quality and annotated 3D ground truth data Qian et al. (2023). Although LiDAR technology facilitates data collection, the manual annotation is laborious Wei et al. (2021); Liu et al. (2022b;a); Paat et al. (2024). This underscores the urgent need for efficient 3D automated annotation models.

Obtaining 2D labels is significantly easier compared to 3D annotations. Early works have explored automating the 3D annotation process with weak labels, e.g., 2D bounding box Wei et al. (2021); Qi et al. (2018); Liu et al. (2022b;a); Qian et al. (2023); Paat et al. (2024); Huang et al. (2023a), center-clicks Meng et al. (2020; 2021a), and 2D segmentation labels McCraith et al. (2021); Wilson et al. (2020). These studies aim to use weak 2D annotations to infer precise 3D labels, employing advanced algorithms to discern objects from the background in 3D space to ensure quality. However, existing techniques struggle with the inherent sparsity and irregularity of point clouds, especially in annotating truncated, occluded, or distant hard samples. This highlights the necessity for approaches that effectively utilize multimodal information to complement point cloud sparsity in challenging scenarios for 3D automatic annotations.

To this end, multimodal models Liu et al. (2022a); Paat et al. (2024); Liu et al. (2022b) have been developed to leverage imagery and point clouds, extracting and combining features from both to enhance annotation accuracy. Despite promising performance on benchmarks like KITTI Geiger et al. (2012), challenges persist in accurately annotating hard samples. The primary challenge arises from the lossy nature of camera-to-LiDAR and LiDAR-to-camera projections, where only 5% of

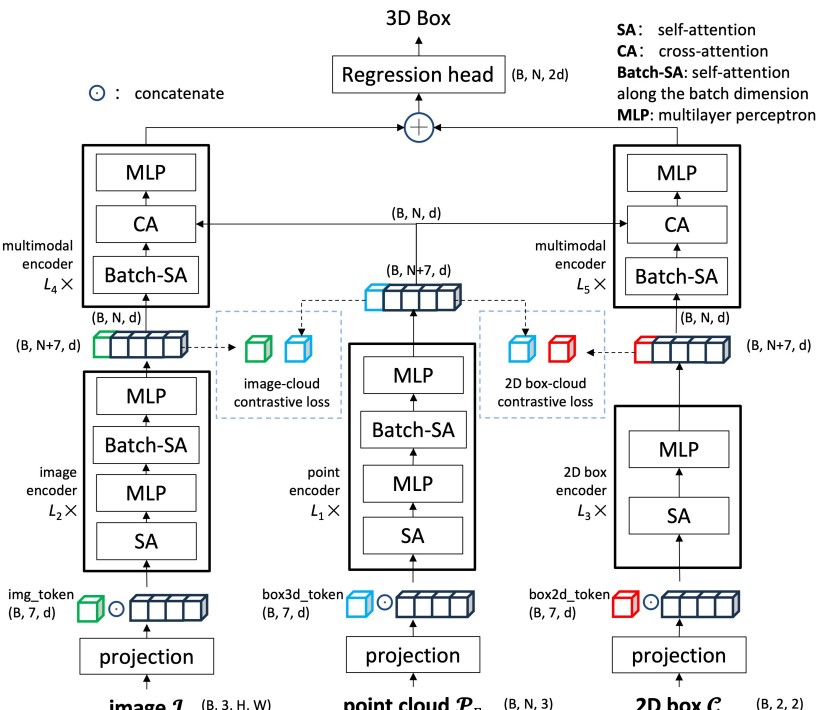

Figure 1: Illustration of MMCAT: Integrates images and 2D boxes with 3D annotations using advanced encoders. The image encoder extracts features and aligns learnable $img\_tokens$ with $box3d\_tokens$ from point clouds, refined by SA and Batch-SA. The 2D box encoder inputs bounding embeddings along learnable $box2d\_tokens$, refined by SA. They are under contrastive optimization for alignment with point features from the point encoder. Fusion features are unified in two multimodal encoders using Batch-SA and CA, resulting in accurate 3D bounding boxes.

visual data matches LiDAR points Liu et al. (2022c), and depth information from point clouds is inadequately retained. This emphasizes the necessity for an effective multimodal approach to reconciling discrepancies between image and point cloud data. Combining the semantic details from images with the geometric data from point clouds can significantly improve annotation accuracy, especially for hard samples.

Inspired by the success of text-image frameworks Radford et al. (2021); Li et al. (2022; 2021), we extend this approach to automatic point cloud annotation using. We introduce MMCAT, a multimodal context-aware transformer, for automated 3D point cloud annotation with 2D visual guidance. MMCAT employs specialized encoders for point clouds, images, and 2D boxes to extract and align features across 2D and 3D domains. MMCAT's point encoder captures the geometric structure of point clouds, while the image encoder extracts dense semantic information from 2D images. Additionally, a 2D box encoder is designed to align 2D box coordinates with 3D point clouds, enhancing their geometry with spatial context from images. This is particularly useful for challenging samples with limited clarity. These encoders effectively integrate geometric and semantic data within a contrastive embedding space. Guidance from images and 2D boxes in multimodal encodes supplements sparse point data, offering a more comprehensive understanding of 3D representations and improving our model's accuracy in 3D box generation. As shown in Figure 1, this integration is crucial for handling hard samples and achieving precise 3D annotations. In summary, this paper makes the following contributions:

1. We introduce MMCAT, a multimodal framework where visual cues from images and 2D boxes enhance 3D point cloud representation, thereby improving 3D annotations.

2. We develop specialized encoders featuring a batch-attention mechanism to enhance interaction among point cloud, image, and 2D box features, markedly boosting multimodal data alignment and integration.

3. MMCAT excels in improving 3D representations for hard samples with 2D visuals, providing accurate annotations where conventional methods often struggle.

4. MMCAT achieves SOTA performance on the KITTI dataset and pioneers in assessing its capabilities on the Waymo Open dataset, demonstrating its adaptability and effectiveness across varied data environments.

## 2 RELATED WORK

### 2.1 3D REPRESENTATION LEARNING

Traditional 3D point cloud techniques, like voxel-based networks Maturana & Scherer (2015); Wu et al. (2015); Xie et al. (2018); Yang et al. (2018) and point-based methods Qi et al. (2017a;b); Lang et al. (2019); Shi et al. (2019); Qi et al. (2018), have shown significant utility but often do not fully exploit the inherent geometric details of point clouds. Recently, attention-based models Zhao et al. (2021); Yu et al. (2021); Guo et al. (2021); Liu et al. (2022a;b); Qian et al. (2023), especially those utilizing Transformer architectures, have emerged as powerful tools in 3D vision, overcoming previous limitations by efficiently processing point cloud data. These models, including the pioneering Point Transformer Zhao et al. (2021) and its variants Wu et al. (2022); Park et al. (2022), are tailored to meet point cloud data's unique requirements, achieving notable success in object detection Misra et al. (2021); Pan et al. (2021); Park et al. (2022), segmentation Zhao et al. (2021); Wu et al. (2022); Lai et al. (2022); Park et al. (2022), classification Liu et al. (2020); Qi et al. (2017a;b), and annotation Liu et al. (2022a;b); Qian et al. (2023); Paat et al. (2024). Their ability to capture long-range dependencies and facilitate improved information exchange has led to breakthroughs in these areas.

Nevertheless, the challenge of harnessing 3D point cloud data's full potential persists, especially with sparse and irregular hard-to-detect samples. Although point clouds provide detailed geometry, they often miss semantic context essential for comprehensive scene understanding. Images are crucial for complementing 3D representations with their dense semantic information. Yet, many current multimodal fusion methods Liu et al. (2022b;a); Paat et al. (2024) experience substantial loss of geometric and semantic information during the integration into 3D point clouds.

Inspired by the CLIP architectures' success Radford et al. (2021); Li et al. (2021; 2022); Kim et al. (2021), which adeptly connect text and image modalities via text-image pairs, we propose its extension to image-point pairs for 3D point cloud representation learning. Our method utilizes visual guidance (images) to enrich 3D representation learning, effectively overcoming the shortcomings of existing multimodal fusion autolabelers. This method employs 2D visual guidance to extract and improve 3D representations, significantly enhancing the accuracy and reliability of 3D data.

### 2.2 3D ANNOTATION FROM WEAK LABELS

Human annotation of point cloud data is an arduous task that has spurred the development of automated techniques using weak labels Meng et al. (2021a); Wei et al. (2021); Liu et al. (2022a;b). The affordability and ease of obtaining 2D annotations compared to 3D labels have spotlighted their potential in automating the annotation process. Current methods leveraging 2D annotations Zakharov et al. (2020); Wei et al. (2021); Liu et al. (2022a); Qian et al. (2023) often face challenges in translating to 3D, hindered by the sparsity and irregularity of point cloud data, particularly with hard samples. While multimodal fusion methods Liu et al. (2022b;a); Paat et al. (2024) for 3D automatic annotation exist, they frequently encounter geometric and semantic losses during the fusion into point clouds.

Expanding on foundational research, our study adapts the text-image multimodal framework into an image-point system for 3D point cloud annotation. We leverage 2D visual hints to enhance point cloud representation, addressing the limitations of existing fusion methods and boosting 3D annotation accuracy. MMCAT efficiently integrates multimodal data, capitalizing on images' semantic richness to improve sparse geometric point data.

## 3 MMCAT FOR 3D POINT CLOUD AUTOMATIC ANNOTATION

This section outlines our methodology, starting with preparation for tri-modal inputs: frustum point clouds, corresponding images, and 2D boxes (Sec. 3.1). We then detail MMCAT architecture, including point cloud, image, and 2D box encoders, along with multimodal fusion encoders (Sec. 3.2). Next, we explore 2D visual guidance (Sec. 3.3) and conclude with our training objectives (Sec. 3.4).

## 3.1 DATA PREPARATION

Our methodology generates 3D pseudo labels for training standard 3D object detectors, using LiDAR point clouds $\mathcal{P}$, corresponding images $\mathcal{I}$, and weak 2D bounding boxes $\mathcal{C}$. Initially, we identify the frustum areas defined by 2D bounding boxes, following prior research Qi et al. (2018); Wei et al. (2021); Liu et al. (2022a;b). With a calibration matrix linking LiDAR and image data, we project 3D point cloud coordinates $(x, y, z)$ onto a 2D image plane, obtaining coordinates $(u, v)$ through the calibration mapping function $f_{\text{cal}}$. This step is crucial for aligning multimodal data sources for effective 3D annotation.

We delineate the 2D projected subset of the point cloud, denoted as $\mathcal{P}_{2D}$, which resides within the confines of a given 2D box $\mathcal{B}$. Mathematically, $\mathcal{P}_{2D} \in \mathbb{R}^{N \times 2}$ is defined as:

$$\mathcal{P}_{2D} = \{(u, v) \mid (u, v) = f_{cal}(x, y, z), \forall (x, y, z) \in \mathcal{P}, (u, v) \in \mathcal{B}\}. \tag{1}$$

wherein $(x, y, z)$ constitutes the coordinates of a point in the 3D space, and $\mathcal{B}$ signifies the delineated region within the 2D bounding box.

Subsequently, the corresponding frustum sub-cloud, $\mathcal{P}_F$, encapsulating the 3D points that project within the 2D bounding region, is extracted and formalized as a subset of $\mathbb{R}^{N \times 3}$, expressed as:

$$\mathcal{P}_F = \{(x, y, z) \mid f_{cal}(x, y, z) \in \mathcal{P}_{2D}\}, \tag{2}$$

The variable $N$ represents the number of points in each input frustum sub-cloud $\mathcal{P}_F$ within each batch. We combine $\mathcal{P}_F$ with corresponding images $\mathcal{I}$ and 2D bounding box coordinates $\mathcal{C}$. Following ViT Dosovitskiy et al. (2020), images $\mathcal{I}$ are processed into $k \times k$ patches. For fusing 2D visual cues with 3D data, 2D box coordinates $\mathcal{C}$ are set within the scene's coordinate system as $(l, t, r, b)$ to outline the box's edges. We use 2D ground truth as it is significantly easier to obtain than 3D annotations. The tri-modal inputs integrate geometric details from point clouds with 2D imagery to enrich 3D understanding.

## 3.2 MMCAT ARCHITECTURE

The MMCAT architecture processes tri-modal inputs: frustum sub-clouds $\mathcal{P}_F$ in $(B, N, 3)$, image data $\mathcal{I}$ in $(B, 3, W, H)$, and 2D boxes $\mathcal{C}$ in $(B, 2, 2)$, to output 3D boxes $(B, 7)$. To address the variable density of point clouds from LiDAR scans, we use random sampling to standardize point counts within a batch, aligning them to the batch's median, thus handling point density variations.

MMCAT integrates point cloud, image, and 2D box encoders, followed by two multimodal encoders. The point encoder transforms $\mathcal{P}_F$ into $d$-dimensional embeddings using projection layers, resulting in $(B, N, d)$ shapes. Image data ($k \times k$-sized patches) are projected into embeddings of shape $(B, p, d)$ by an MLP, where $p$ represents the patch count. The 2D box encoder similarly converts boxes into embeddings with $(B, 2, d)$ dimensions. This standardizes tri-modal inputs for fusion, allowing accurate feature extraction and alignment. MMCAT's encoders capture features from each modality and effectively integrate their unique contributions. Two multimodal encoders then refine interactions between (image, point) and (box, point) pairs, focusing on 3D label generation. For detailed insights into visual guidance, refer to Sec. 3.3. Figure 1 illustrates the MMCAT architecture for 3D annotation.

**Point Encoder.** Our point encoder features a custom block for extracting intra- and inter-object features from 3D point clouds. This block sequence starts with a pre-norm layer, proceeds with self-attention (SA) and a multilayer perceptron (MLP), incorporates a batch self-attention (Batch-SA), and ends with an additional MLP. Unlike traditional SA, which works on the sequence dimension, Batch-SA extends to the batch dimension, capturing essential inter-object relations for precise 3D scene representation, especially informative for complex samples. As depicted in Figure 1, the encoder utilizes $L_1$ blocks to enhance point embeddings from input projections. To enhance alignment across different modalities in 3D, we introduce seven learnable 3D box tokens, denoted as $box3d\_tokens$, supplemented by point embeddings. These tokens encode the essential attributes of a 3D bounding box: location $(x, y, z)$, dimensions $(l, w, h)$, and $yaw$ angle. These attributes are crucial to accurately define the position, size, and orientation of the 3D box in a 3D context, hence leading to a total of seven tokens. This enables each block to generate 3D features as $(B, N + 7, d)$, with $box3d\_tokens$ aligning with image and 2D box tokens via contrastive learning for accurate alignment.

**Image Encoder.** Our image encoder connects 3D point clouds with 2D visuals, converting image data into a format optimized for multimodal fusion. Image patches $(B, p, k, k)$ are transformed into

visual embeddings $(B, p, d)$ through an MLP projection. We match $N$ 3D points to 2D image patches based on their 2D coordinates, creating a $(B, N, d)$ matrix of 2D features for corresponding 3D points. The encoder, tailored for image data, mirrors the point encoder's design through $L_2$ stacked blocks, consisting of a pre-norm layer, SA, an MLP, Batch-SA, and a concluding MLP (see Figure 1). This uniform architecture across modalities eliminates the need for different designs, ensuring consistency. The image encoder leverages SA and Batch-SA to extract 2D intra- and inter-object semantic visual features in $(B, N, d)$. Concurrently, we engage in contrastive learning with $box3d\_tokens$ from the point encoder in $(B, 7, d)$, employing seven dimensioned tokens, $img\_tokens$ in $(B, 7, d)$. Contrastive loss optimization aligns these tokens with point cloud features, resulting in 2D image representation in $(B, N + 7, d)$. This strategic alignment is fundamental to our methodology, enabling accurate 3D analysis by integrating visual cues with 3D.

**2D Box Encoder.** Our approach improves 3D spatial perception with a 2D box encoder that derives spatial insights from 2D coordinates, offering additional visual cues for point clouds. This method links 2D and 3D data, facilitating the creation of 3D boxes from 2D outlines. Equipped with SA and MLP blocks within a transformer architecture, the 2D box encoder processes features into $(B, 2 + 7, d)$ format through $L_3$ stacked blocks, aligning with $box3d\_tokens$ from point clouds as shown in Figure 1. Similarly, we use seven tokens to represent $box2d\_tokens$, facilitating contrastive learning with $box3d\_tokens$. Incorporating 2D coordinates into our framework provides enhanced visual guidance by clearly defining spatial boundaries and identifying object locations, greatly improving the precision of 3D object interpretation. This is particularly beneficial for hard samples with occlusions and truncations.

**Multimodal Encoder.** Our multimodal encoding framework merges point cloud, image, and 2D box features through transformer blocks, utilizing pre-norm, Batch-SA, cross-attention (CA), and MLP layers to unify modalities and enhance 3D object representation. For image-point fusion, a multimodal encoder refines point features $(B, N, d)$ using Batch-SA and CA, integrating dense image semantics $(B, N, d)$ into point cloud data. This boosts 3D representation learning, crucial for precise 3D box regression in complex scenarios. Similarly, for 2D box-point integration, spatial constraints from 2D boxes $(B, N, d)$ are processed with point features in another multimodal encoder, enhancing $(B, N, d)$ 3D point features. We upscale the 2D box encoder output from $(B, 2, d)$ to $(B, N, d)$ via an MLP to facilitate enhanced fusion. Concatenated fused features $(B, N, 2d)$ from both encoders prepare the data for precise 3D box regression. This method leverages 2D guidance to deepen 3D scene understanding, merging visual cues with spatial insights for comprehensive 3D representation.

## 3.3 VISUAL GUIDANCE FROM 2D

Even with 3D human annotations, robust 3D point cloud representation remains challenging due to their often incomplete, sparse, and noisy nature Huang et al. (2023b); Wei et al. (2021), stemming from LiDAR's inherent limitations. To address this, we propose enriching 3D representation learning with 2D dense visual hints. In MMCAT, images and 2D box features serve as contexts $(K, V)$ in dual multimodal encoders, implementing two visual guidance strategies: (image, point) and (box, point).

**Visual Guidance from (image, point) Modalities.** In the image-point process, images provide dense visual contexts, offering semantic details like color and texture absent in point cloud data. The CA in our multimodal encoder enhances the interaction between 2D images and point cloud features, enriching point cloud descriptions with semantic details and resolving ambiguities. For instance, image guidance can help differentiate objects with similar geometric shapes but distinct appearances in the image domain. This guidance forms more effective 3D point cloud representations, particularly for hard samples, where image semantics can significantly enhance sparse point cloud representation. Integrating 2D and 3D information improves the representation of hard samples, highlighting the importance of multimodal integration for accurate and reliable 3D annotations.

**Visual Guidance from (box, point) Modalities.** In the (box, point) modality, 2D bounding boxes offer localized visual cues, highlighting areas of interest in the point cloud. As spatial constraints in our multimodal encoder, these features link 2D visuals with 3D spatial data. For example, a 2D box around a vehicle in an image helps the model focus on refining the vehicle's point cloud representation, crucial for obscured hard samples. The 2D box outlines the entire vehicle, including occluded sections, offering complete size details. This allows for a more accurate 3D box, even without point cloud data for hidden parts. This approach enriches point cloud geometry with precise spatial details from 2D boxes, resulting in more accurate localization and sizing of 3D objects.

Table 1: Results of KITTI official *test* set (Vehicle), compared to the fully supervised PointRCNN and other weakly supervised baselines.

| Method | Modality | Full Supervision | $AP_{3D}(IoU = 0.7)$ | | | $AP_{BEV}(IoU = 0.7)$ | | |
|---|---|---|---|---|---|---|---|---|
| | | | Easy | Moderate | Hard | Easy | Moderate | Hard |
| PointRCNN Shi et al. (2019) | LiDAR | ✓ | 86.96 | 75.64 | 70.70 | 92.13 | 87.39 | 82.72 |
| MV3D Chen et al. (2017) | LiDAR | ✓ | 74.97 | 63.63 | 54.00 | 86.62 | 78.93 | 69.80 |
| F-PointNet Qi et al. (2018) | LiDAR | ✓ | 82.19 | 69.79 | 60.59 | 91.17 | 84.67 | 74.77 |
| AVOD Ku et al. (2018) | LiDAR | ✓ | 83.07 | 71.76 | 65.73 | 90.99 | 84.82 | 79.62 |
| SECOND Yan et al. (2018) | LiDAR | ✓ | 83.34 | 72.55 | 65.82 | 89.39 | 83.77 | 78.59 |
| PointPillars Lang et al. (2019) | LiDAR | ✓ | 82.58 | 74.31 | 68.99 | 90.07 | 86.56 | 82.81 |
| SegVoxelNet Yi et al. (2020) | LiDAR | ✓ | 86.04 | 76.13 | 70.76 | 91.62 | 86.37 | 83.04 |
| Part-$A^2$ Shi et al. (2020b) | LiDAR | ✓ | 87.81 | 78.49 | 73.51 | 91.70 | 87.79 | 84.61 |
| PV-RCNN Shi et al. (2020a) | LiDAR | ✓ | 90.25 | 81.43 | 76.82 | 94.98 | 90.65 | 86.14 |
| Comparison with other Autolabelers (PointRCNN) | | | | | | | | |
| WS3D Meng et al. (2020) | LiDAR | BEV Centroid | 80.15 | 69.64 | 63.71 | 90.11 | 84.02 | 76.97 |
| WS3D(2021) Meng et al. (2021a) | LiDAR | BEV Centroid | 80.99 | 70.59 | 64.23 | 90.96 | 84.93 | 77.96 |
| FGR Wei et al. (2021) | LiDAR | 2D Box | 80.26 | 68.47 | 61.57 | 90.64 | 82.67 | 75.46 |
| MAP-Gen Liu et al. (2022b) | LiDAR+RGB | 2D Box | 81.51 | 74.14 | 67.55 | 90.61 | 85.91 | 80.58 |
| MTrans Liu et al. (2022a) | LiDAR+RGB | 2D Box | 83.42 | 75.07 | 68.26 | 91.42 | 85.96 | 78.82 |
| CAT Qian et al. (2023) | LiDAR | 2D Box | 84.84 | 75.22 | 70.05 | 91.48 | 86.77 | 79.93 |
| MED-LU Paat et al. (2024) | LiDAR+RGB | 2D Box | 85.49 | 75.96 | 69.12 | 91.86 | 86.68 | 79.44 |
| MMCAT (ours) | LiDAR+RGB | 2D Box | 85.44 | **77.02** | **71.08** | 91.62 | **87.17** | **81.08** |

Visual guidance from 2D modalities serves a dual role in multimodal encoders. Images provide semantic guidance, enriching 3D representations, while 2D boxes offer spatial guidance, ensuring precise localization and dimensioning of objects in 3D. Together, they enable a semantically richer interpretation of 3D point clouds, especially for challenging samples, leveraging the strengths of each modality for robust point cloud interpretation.

## 3.4 Loss Function

Our MMCAT model's training is enhanced by a composite loss function that includes distance-IoU (dIoU) loss ($\mathcal{L}_{box}$), directional loss ($\mathcal{L}_{dir}$), and contrastive learning losses ($\mathcal{L}_{ip}$ for image-point and $\mathcal{L}_{bp}$ for box-point). Here dIoU Loss ($\mathcal{L}_{box}$) measures the discrepancy between the predicted box and the ground truth. This metric ensures that predicted 3D boxes precisely represent the objects within the point cloud. Directional Loss ($\mathcal{L}_{dir}$) tackles IoU's direction-invariance by categorizing orientations into the front ($[-\pi/2, \pi/2)$) and back ($[\pi/2, \pi] \cup [-\pi, -\pi/2)$), using an MLP for accurate 3D box orientation detection. The directional loss is calculated using cross-entropy. For contrastive learning losses, we define $\mathcal{L}_{ip}$ for image-point alignment and $\mathcal{L}_{bp}$ for box-point alignment to refine unimodal representations before their fusion. These losses are designed to optimize a cosine similarity function, enhancing the similarity scores for congruent pairs of image-point and box-point tokens. The $\mathcal{L}_{ip}$ and $\mathcal{L}_{bp}$ follow the contrastive loss functions used in Radford et al. (2021); Li et al. (2021). In our approach, the contrastive loss is computed over the entire batch. For a batch size of $B$, we compute the cosine similarity between all (point, image) and (point, box2d) token pairs, resulting in an $B \times B$ similarity matrix. The $img\_token$, $box2d\_token$, and $box3d\_token$ in $(B, 7, d)$ represent linear transformations mapping trainable tokens to normalized, lower-dimensional representations for the image, 2D box, and point cloud, respectively. Each token contributes to each sample, as deep features. They are learned through the contrastive learning process. The total loss function is formulated as follows:

$$\mathcal{L} = \lambda_{box}\mathcal{L}_{box} + \mathcal{L}_{ip} + \mathcal{L}_{bp} + \mathcal{L}_{dir}, \tag{3}$$

with $\lambda_{box}$ set to 5, it is balancing the contribution of each component based on empirical findings Qian et al. (2023); Liu et al. (2022a).

## 4 Experiments

### 4.1 Experimental Setup

We assessed MMCAT's performance on the KITTI and Waymo Open Datasets using five transformer-based encoders: point, image, 2D box, and multimodal encoders. We also conducted ablation studies to highlight the distinct impact of each module on MMCAT's effectiveness.

Table 2: Results of KITTI *val* set (Vehicle), compared to the fully supervised PointRCNN and other weakly supervised baselines.

| Method | Modality | Full Supervision | $\text{AP}_{3D}(IoU = 0.7)$ | | |
| --- | --- | --- | --- | --- | --- |
| | | | Easy | Moderate | Hard |
| PointRCNN Shi et al. (2019) | LiDAR | ✓ | 88.88 | 78.63 | 77.38 |
| WS3D Meng et al. (2020) | LiDAR | BEV Centroid | 84.04 | 75.10 | 73.29 |
| WS3D(2021) Meng et al. (2021a) | LiDAR | BEV Centroid | 85.04 | 75.94 | 74.38 |
| FGR Wei et al. (2021) | LiDAR | 2D Box | 86.68 | 73.55 | 67.91 |
| MAP-Gen Liu et al. (2022b) | LiDAR+RGB | 2D Box | 87.87 | 77.98 | 76.18 |
| MTrans Liu et al. (2022a) | LiDAR+RGB | 2D Box | 88.72 | 78.84 | 77.43 |
| CAT Qian et al. (2023) | LiDAR | 2D Box | 89.19 | 79.02 | 77.74 |
| MED-LU Paat et al. (2024) | LiDAR+RGB | 2D Box | 89.26 | 75.27 | 78.05 |
| MMCAT (ours) | LiDAR+RGB | 2D Box | **89.43** | **79.10** | **79.27** |

Table 3: Accuracy comparison of KITTI training set (Vehicle) annotations across various difficulties.

| | Hard (mIoU) | Moderate (mIoU) | Easy (mIoU) |
| --- | --- | --- | --- |
| Mtran Liu et al. (2022a) | 80.70 | 85.86 | 89.09 |
| CAT Qian et al. (2023) | 83.02 | 86.66 | 89.88 |
| MMCAT | **86.19** | **88.25** | **91.14** |

**KITTI Dataset.** Our evaluation on the KITTI dataset Geiger et al. (2012) involved 3,712 training frames with 15,654 vehicle instances and 3,769 validation frames. Following standard protocols, we trained on 500 frames and validated on 3,769 frames for consistency with benchmarks. We assessed MMCAT's performance using PointRCNN across Easy, Moderate, and Hard difficulty levels based on the official KITTI evaluation metrics. Performance metrics included mean Average Precision (mAP) at a 0.7 IoU threshold for 3D and Bird's Eye View (BEV) detection. Aligned with recent 3D annotation research Wei et al. (2021); Liu et al. (2022a); Qian et al. (2023); Paat et al. (2024), our analysis focused on the Vehicle category and excluded objects with fewer than 15 foreground points to ensure feasible 3D bounding box annotations.

**Waymo Open Datset.** The Waymo Open Dataset Sun et al. (2020), with 798 training and 202 validation sequences for vehicles, is recorded by a 64-channel LiDAR, capturing around 180,000 points every 0.1 seconds. Our evaluation employs two primary 3D object detection metrics: mAP for bounding box accuracy and mAP weighted by Heading Accuracy (mAPH), factoring in object orientation, with IoU thresholds set at 0.7 for vehicles. Detection difficulty is classified into LEVEL_1 (boxes with $> 5$ LiDAR points) and LEVEL_2 (boxes with $\geq 1$ LiDAR point). We conducted experiments on version 1.2.0 of the Waymo and focused exclusively on the Vehicle class.

**Implementation Details.** MMCAT was developed in PyTorch Paszke et al. (2019), using customized Transformer encoder layers for point cloud, image, 2D box, and two multimodal encoders, as illustrated in Figure 1. The design of MMCAT's image and multimodal encoders was inspired by CLIP Radford et al. (2021) and ALBEF Li et al. (2021), while the point encoder was based on CAT Qian et al. (2023), aiming to balance accuracy with minimal complexity. The point encoder includes $L_1 = 3$ blocks with SA, MLP, Batch-SA, and an MLP featuring two linear layers with ReLU activations. The image encoder also has $L_2 = 3$ blocks, the 2D box encoder $L_3 = 2$, and each multimodal encoder $L_4 = L_5 = 2$ blocks, with SA, CA, and Batch-SA configured to a hidden size of $d = 512$ and 8 attention heads. Box regression is achieved through three two-layer MLPs with a hidden size of 1024. We used the Adam optimizer with a starting learning rate of $2 \times 10^{-4}$, a cosine annealing scheduler for adjustments, and a weight decay of 0.05. MMCAT trained on four Nvidia A100 GPUs for 1,000 epochs with a batch size of $B = 256$, incorporating standard data augmentations like shifting, scaling, and flipping. We employed the frustum extraction method Wei et al. (2021); Liu et al. (2022a); Qian et al. (2023); Meng et al. (2021b) to extract point clouds and their associated images and 2D boxes for the Car category from all frames or sequences in the datasets. These extracted objects were randomly selected to train our model in 14% and 20% for KITTI and Waymo, respectively. Once trained, MMCAT serves as a 3D automatic annotator, re-labeling the KITTI and Waymo training set for the Vehicle class. For 3D object detection, we utilized PointRCNN for KITTI and PVRCNN++ Shi et al. (2023) for Waymo, following the OpenPCDET Team (2020) protocols for both training and evaluation.

Table 4: Results of Waymo *val* set (Vehicle), compared to fully-supervised PVRCNN++. We show mAP and mAPH for Level 1 and 2 Vehicles.

| Method | Modality | Full Supervision | Level 1 | | Level 2 | |
|---|---|---|---|---|---|---|
| | | | mAP | mAPH | mAP | mAPH |
| PVRCNN++ Shi et al. (2019) | LiDAR | ✓ | 79.25 | 78.78 | 70.61 | 70.18 |
| MTrans Liu et al. (2022a) | LiDAR+RGB | 2D Box | 70.12 | 70.66 | 61.80 | 63.59 |
| MMCAT (ours) | LiDAR+RGB | 2D Box | 72.81 | 72.43 | 65.62 | 65.53 |

Table 5: MMCAT generated 3D boxes in comparison with Waymo human annotations. We show mIoU, Recall with an IoU threshold of 0.7, and Recall with a location error (LE) threshold of 0.5.

| | mIoU↑ | Recall (IoU = 0.7) | Recall (LE = 0.5) |
|---|---|---|---|
| MMCAT (train) | 72.89 | 73.61 | 89.51 |
| MMCAT (val) | 70.06 | 70.92 | 88.14 |

## 4.2 MAIN RESULTS AND COMPARISON WITH SOTAS

**Quantitative Analysis on KITTI Set (Vehicle).** Submitting MMCAT-trained PointRCNN results to the KITTI evaluation server showcased our approach's competitiveness with fully supervised methods, using just 500 labeled frames instead of the entire 3,712 scenes with 15,654 vehicle instances. MMCAT's pseudo labels allowed PointRCNN to reach 99% performance of its manually annotated counterpart in Table 1. Our $AP_{3D}$ closely aligns with that of PointRCNN trained on human annotations, marking a six-fold reduction in extensive manual labeling for the KITTI vehicle.

MMCAT sets a new SOTA in 3D annotation, outperforming current SOTAs in Moderate and Hard tasks with improvements of 1.96% and 1.06% compared to the latest multimodal model MED-LU, respectively. On the KITTI validation set in Table 2, MMCAT-trained PointRCNN matches the performance of its self-supervised variant. In challenging scenarios, MMCAT distinguishes itself by leveraging its distinctive capabilities in Hard and Moderate tasks in Table 1 and 2. Table 3 presents a comparative analysis of annotation accuracy on samples of varying difficulty between MMCAT and current SOTAs Qian et al. (2023); Liu et al. (2022a).

It is observed that MMCAT outperforms the current multimodal model MTrans Liu et al. (2022a) in annotating hard samples, with an improvement of around 3% in IoU. Despite our model being 15% larger compared to the CAT model, our inference time remains the same as CAT's. For detailed information, please refer to the **Appendix** B.3. In our research, we have also expanded our experiments to include the pedestrian category. The MMCAT model has achieved excellent annotation results across these additional tests. For more detailed information, please refer to **Appendix** B.1 and B.2 in the supplementary materials.

**Quantitative Analysis on Waymo Set (Vehicle).** MMCAT's performance on the Waymo verifies that it achieves comparable detection accuracy with significantly less labeled data. Utilizing only 20% labeled data ($400,000$ vehicles), MMCAT approaches the performance of PVRCNN++ trained on the whole dataset ($2,445,159$ vehicles). On the Waymo validation set in Table 4, MMCAT's pseudo labels enable PVRCNN++ to reach mAP and mAPH scores close to those of a fully supervised model, 72.81% mAP and 72.43% mAPH for Level 1 vehicles, and 64.62% mAP and 64.53% mAPH for Level 2 vehicles. It demonstrates 91.8% of the fully supervised performance.

MMCAT's annotation efficiency is further highlighted to assess class-specific impacts through direct comparisons with manual ground truth labels. The results in Table 5 achieved a mean IoU of 72.89% and a recall rate of 73.61%. Evaluation metrics include mIoU, Recall with an IoU threshold of 0.7, and Recall with an LE threshold of 0.5. It outperforms the latest multimodal model MTrans with around 3% and 4% mAP in Level 1 and Level 1 vehicles. This learning enables the model to comprehensively relabel the whole training set of Waymo Vehicle, effectively handling the $80\%$ of unseen data while utilizing $20\%$ data for training. This suggests that MMCAT could serve as a 3D automatic annotator for Waymo, significantly reducing 80% of time-consuming and labor-intensive human annotation costs during data preparation, thus underlining the generalization capability of our framework.

**Qualitative Analysis.** Figure 2 showcases MMCAT's pseudo-labeling prowess on the KITTI training set against ground truth (blue boxes), particularly with hard samples marked by sparse point

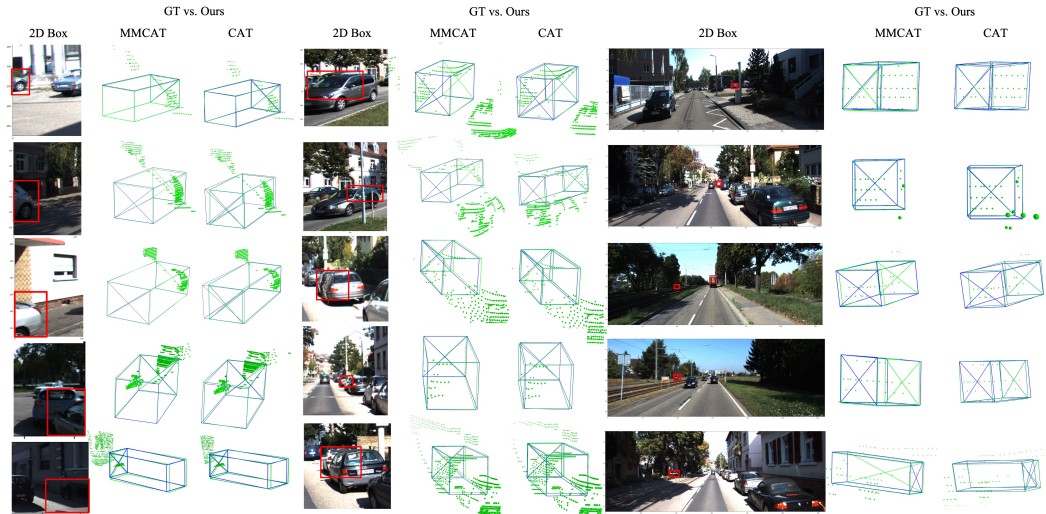

Figure 2: Qualitative Analysis with MMCAT annotating KITTI training Set: MMCAT exhibits significant robustness and precision in generating 3D bounding boxes, particularly for challenging samples. This includes significantly truncated objects (first two columns), heavily occluded (middle two columns), and far from the sensor (last two columns). Under these conditions, MMCAT produces amodal bounding boxes (in green) that accurately encompass the entire vehicle structure.

Table 6: Ablation results on KITTI *val* split (Vehicle).

| | Point enc. | Image enc. | 2D Box enc. | Multimodal enc. | Batch-SA | mIoU | Recall | mAP | mAP$_{R40}$ |
|---|---|---|---|---|---|---|---|---|---|
| Model A | ✓ | | | | | 63.11 | 44.50 | 46.31 | 49.24 |
| Model B | ✓ | ✓ | | | | 68.11 | 63.01 | 68.31 | 69.09 |
| Model C | ✓ | ✓ | ✓ | | | 71.11 | 68.01 | 75.10 | 77.99 |
| Model D | ✓ | ✓ | ✓ | | ✓ | 73.28 | 73.94 | 80.33 | 83.72 |
| Model E | ✓ | ✓ | ✓ | ✓ | | 77.13 | 77.96 | 86.01 | 87.34 |
| **Ours** (500) | ✓ | ✓ | ✓ | ✓ | ✓ | 78.03 | 78.34 | 88.52 | 90.69 |
| **Ours** (all) | ✓ | ✓ | ✓ | ✓ | ✓ | 83.32 | 88.21 | 93.66 | 95.68 |

distribution. MMCAT generates accurate bounding boxes in challenging conditions, including heavy truncation (first two columns), occlusion (middle two columns), or significant distance from the sensor (last two columns). This performance is driven by enriching 3D point cloud representations with 2D visual cues and the capture of inter-object relations, refining initially incomplete hard samples. These examples underscore MMCAT's effectiveness and the utility of combining 2D and 3D inputs for automated annotations, allowing for accurate 3D annotations in diverse scene settings.

## 4.3 ABLATION STUDIES

We conducted ablation studies to evaluate the individual impact of MMCAT's components on model performance. These components include the point, image, and 2D box encoders for processing different data types, Batch-SA for inter-object relation learning, and multimodal encoders for integrating modalities. Performance was assessed using mIoU, recall at an IoU threshold of 0.7, mAP, and mAP at 40 recall points (mAP$_{R40}$) within the Vehicle category. This analysis allows us to pinpoint the contribution of each component to MMCAT's effectiveness in generating accurate 3D annotations.

Table 6 outlines our ablation study findings, with the standard Transformer encoder as the baseline for our point encoder (Model A). Our analysis reveals: **Model A** (Point Encoder): Extracts geometric data from point clouds, setting the foundation for subsequent enhancements. **Model B** (Image-Point Encoders): Incorporates visual guidance results in the performance boosts of 5%, 18.51%, 22%, and 19.85% on mIoU, Recall, mAP, and mAP$_{R40}$, highlighting the benefit of image semantics for enhancing 3D representations. **Model C** (Image-Point + Box-Point Encoders): Adding a 2D box encoder boosts mIoU by 3%, indicating the value of spatial constraints from 2D boxes in 3D bounding box generation. **Model D** (+Batch-SA): Integrating Batch-SA into Model C increases mIoU by 2.17%,

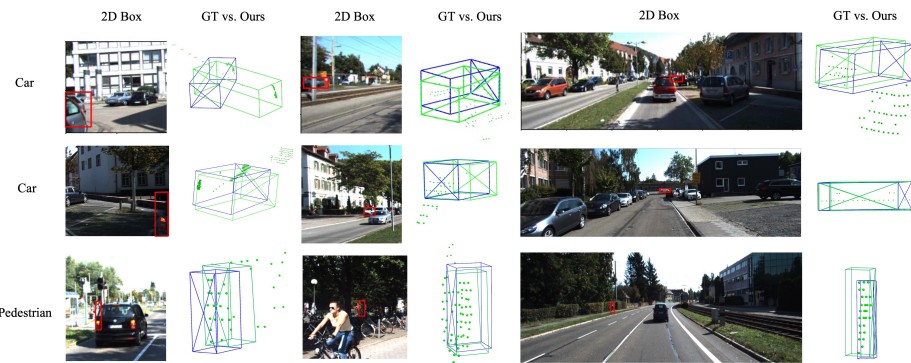

Figure 3: Illustration of failure cases in 3D car and pedestrian annotations on the KITTI *val* set. This figure emphasizes the challenges posed by sparse foreground point clouds (fewer than 30 points) affecting the accuracy of vehicle location, yaw, direction, and pedestrian dimensions in our 3D box predictions (in green). The ground truth boxes are depicted in blue. Zoom in for better clarity.

proving its efficacy in capturing inter-object relations and enhancing sample interactions. **Model E (Comprehensive Multimodal Encoders)**: Finalizing with multimodal encoders for both image-point and box-point modalities yields further 3.85% IoU and 4.02% Recall gains, demonstrating the multimodal encoder's capacity to utilize 2D visual cues for a more integrated 2D and 3D analysis. With the complete MMCAT framework, including Batch-SA, we achieve a remarkable 78.34% Recall, 70.84% mIoU, and 88.52% mAP, confirming the overall model's effectiveness.

Moreover, we evaluate contrastive loss's role in aligning image-point and box-point modalities. Excluding contrastive learning from MMCAT results in a significant decrease of 2.5% in IoU and 3.5% in mAP. This drop underscores the importance of modality alignment for improved fusion and interaction within multimodal encoders, crucial for enhancing 3D box regression accuracy.

### 4.4 LIMITATIONS

Despite the overall accuracy of our box generation, MMCAT faces challenges under certain difficult conditions. Specifically, for samples with sparse foreground points (fewer than 30 points) in Figure 3, the model struggles to accurately capture the data distribution and appearance of these objects. As a result, we excluded these sparsely populated samples from the training process. We observed that MMCAT cannot reliably model objects with such limited data, leading to inaccurate predictions. For instance, MMCAT often misinterprets partial vehicle segments as entire objects and incorrectly positions these segments as the center of objects. These errors significantly impact the accuracy of locations and yaws in our 3D box regressions. We aim to further explore whether incorporating additional multimodal information (e.g., text) and semantic data can enhance the understanding of object shapes in point clouds.

Furthermore, in response to inquiries regarding our model's performance on easier cases within the KITTI dataset, we have conducted a focused analysis of failure modes. Our findings suggest that while the integration of image features generally enhances detection accuracy, it can sometimes lead to the overshadowing of LiDAR data in less complex scenarios. This imbalance occasionally introduces noise which affects performance negatively. We are actively refining our feature integration process to ensure that both modalities contribute optimally and synergistically. This ongoing work underscores our commitment to improving MMCAT and further illustrates the challenges of creating a truly robust multimodal annotation system.

## 5 CONCLUSION

This study presents MMCAT, an autolabeler that generates 3D bounding boxes from weak 2D annotations using LiDAR point clouds and images. Addressing point cloud sparsity, MMCAT employs a multimodal, context-aware Transformer framework that integrates 3D geometric properties, image semantics, and spatial context from 2D boxes, guided by visual cues. Experimental evaluations on KITTI and Waymo datasets confirm MMCAT's superior performance over existing autolabelers, providing high-quality 3D annotations, even for hard samples. Future efforts will focus on enhancing MMCAT's performance, increasing its efficiency and precision across a broader range of classes in 3D automatic annotation tasks.

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
