# Appendices

## A    BACKGROUND OF TRANSFORMER ARCHITECTURE AND SELF-ATTENTION

This section outlines the Transformer architecture and its self-attention mechanism, foundational to Vaswani et al. (2017). Designed initially for NLP tasks, the Transformer employs an encoder-decoder structure to process sequences. The encoder converts an input sequence into continuous representations, which the decoder uses to synthesize an output sequence. Each consists of repeated layers that include self-attention and position-wise feedforward networks. The core of the Transformer's innovative approach lies in its self-attention mechanism, which updates the representation of each sequence element based on the context provided by the entire sequence:

$$\text{Attention}(Q, K, V) = softmax(\frac{QK^T}{\sqrt{d_k}})V \tag{4}$$

where $Q$ (Query), $K$ (Key), and $V$ (Value) matrices are generated through linear transformations of the input's hidden states, with $d_k$ representing the dimensionality of these states for appropriate scaling. This self-attention operation allows each element to dynamically adjust its representation by integrating information weighted by attention scores from the entire sequence, thereby enhancing the model's ability to capture contextual relationships within the data.

## B    KITTI PEDESTRIAN ANNOTATION WITH MMCAT

**KITTI Pedestrian Category.** To further demonstrate the efficacy and generalizability of our MMCAT multimodal framework, we conduct experiments on the Pedestrian class of KITTI. It's important to note that, for a predicted 3D box to be considered acceptable in the Pedestrian category, it must achieve an Intersection over Union (IoU) greater than 0.5 with the ground truth, contrasting with the higher threshold of 0.7 sets for the Vehicle category.

**Data Preparation.** Of the 3,712 training scenes in KITTI, only 951 scenes contain pedestrian labels. Considering the small number of training samples, we randomly choose 25% (515 samples) of 2,257 samples in those scenes. Compared to previously fully supervised PointRCNN, which leverages all 951 exhaustively annotated scenes with 2,257 pedestrian samples, we use far fewer and weak 2D supervisions.

**Implementation Details.** As for sparser Pedestrian point clouds, we adapted from previous studies, normalizing point counts per sample within a batch to the median (around 450) across the batch, addressing point density variability. We adjust the MMCAT encoder for the Pedestrian category to avoid overfitting on these sparser samples. The image encoder also has two blocks ($L_2 = 2$), the 2D box encoder two ($L_3 = 2$), and each multimodal encoder comprises two blocks ($L_4 = L_5 = 2$), with SA and Batch-SA configured to a hidden size of 512 and eight attention heads. During inference, similar to the Vehicle class, we apply MMCAT to our frustum point cloud and use IoU = 0.3. The training process takes 1000 epochs with a batch size of 800 on four Nvidia A100 GPUs. We used the Adam optimizer with a starting learning rate of $1 \times 10^{-4}$, a cosine annealing scheduler for adjustments, and a weight decay of 0.05.

### B.1    EXPERIMENT ANALYSIS ON KITTI VAL (PEDESTRIAN) SET.

Our experiments show that our MMCAT model extends effectively to additional categories, such as Pedestrians. As evidenced in Table 7, MMCAT indicates very promising results in the Pedestrian class, surpassing the majority of existing fully supervised models while utilizing only 25% labeled data under 2D weak supervision. This verifies MMCAT's good generalizability across diverse object classes and efficiency in handling smaller objects.

### B.2    QUALITATIVE ANALYSIS ON KITTI VAL (PEDESTRIAN) SET.

In Figure 4, we present visual examples of MMCAT's performance relabeling the KITTI training set for the Pedestrian category. The first two columns (easy samples) illustrate the model's precision in generating 3D bounding boxes for clear, non-occluded pedestrians at a moderate distance

Table 7: Results of KITTI *Val* set (Pedestrian), compared to the fully supervised PointRCNN and other autolabeler.

| Method | Modality | Full Supervision | $AP_{3D}(IoU = 0.5)$ | | | $AP_{BEV}(IoU = 0.5)$ | | |
|---|---|---|---|---|---|---|---|---|
| | | | Easy | Moderate | Hard | Easy | Moderate | Hard |
| PointRCNN Shi et al. (2019) | LiDAR | ✓ | 63.70 | 69.43 | 58.13 | 68.89 | 63.54 | 57.63 |
| PointPillars Lang et al. (2019) | LiDAR | ✓ | 66.73 | 61.06 | 56.50 | 71.97 | 67.84 | 62.41 |
| Part-A$^2$ Shi et al. (2020b) | LiDAR | ✓ | 70.73 | 64.13 | 57.45 | - | - | - |
| STD Yang et al. (2019) | LiDAR | ✓ | 73.90 | 66.60 | 62.90 | 75.09 | 69.90 | 66.00 |
| VoxelNet Zhou & Tuzel (2018) | LiDAR | ✓ | - | - | - | 70.76 | 62.73 | 55.05 |
| *Comparison with other Autolabeler* | | | | | | | | |
| WS3D Meng et al. (2020) | LiDAR | BEV Centroid | 74.65 | 69.96 | 66.49 | 74.99 | 71.23 | 67.45 |
| CAT Qian et al. (2023) | LiDAR | 2D Box | 75.15 | 70.06 | 67.09 | 74.79 | 71.27 | 66.75 |
| MMCAT (ours) | LiDAR | 2D Box | **76.85** | **72.01** | **70.88** | **76.25** | **73.20** | **70.01** |

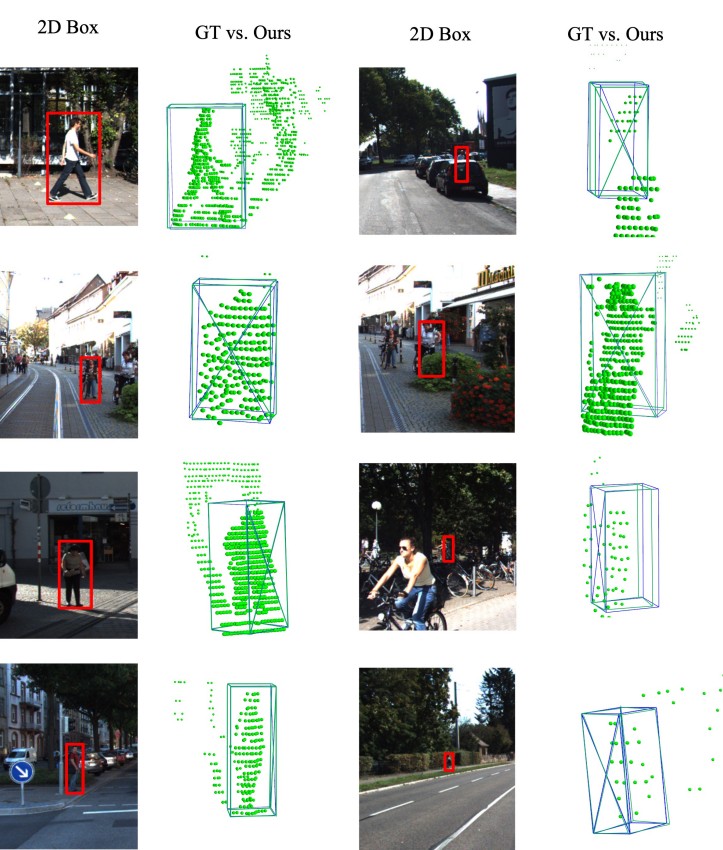

Figure 4: Qualitative Analysis with MMCAT performance on the KITTI Training Set (Pedestrian): Demonstrates MMCAT's robustness and precision in generating 3D bounding boxes, especially under challenging conditions such as substantial truncation, heavy occlusion, or significant distance from the sensor (highlighted in the last two columns). MMCAT effectively generates comprehensive amodal 3D bounding boxes (shown in green) for pedestrian structures.

characterized by dense point clouds. As shown in the last two columns (hard samples), MMCAT also accurately identifies heavily occluded or very far pedestrian samples, significantly sparser than vehicles. This demonstrates MMCAT's capability to process vehicles and adapt to other categories within autonomous driving contexts, even with minimal and more readily available supervision, particularly for challenging samples.

Table 8: Annotation Speed and Training Time.

| Methods | Inference (s) | Training (h) | Parameters (M) |
|---|---|---|---|
| Humman Annotation Song et al. (2015) | 114 | - | - |
| WS3D Meng et al. (2021b) | 2.5 | 10.0 | - |
| MTrans Liu et al. (2022a) | 0.04 | 6.5 | - |
| CAT Qian et al. (2023) | 0.02 | 6.5 | 4 |
| MMCAT (ours) | 0.02 | 7.0 | 4.6 |

B.3 ANNOTATION SPEED.

An analysis of the annotation speed is detailed in Table 8. MMCAT was able to relabel the KITTI training set, which contains 3,712 frames with 15,654 vehicle instances, in about 3 minutes, annotating at 0.02 seconds per instance. In contrast, human annotations take around 114 seconds per instance, or 30 seconds with the assistance of a 3D object detector, as reported in previous studies Huang et al. (2019); Song et al. (2015); Meng et al. (2021b). MMCAT matches CAT's annotation speed despite being a larger model. This trend is consistent on the Waymo, with MMCAT maintaining an annotation speed of 0.02 seconds per car, similar to CAT.