# OpenReview forum: "Multimodal Context-Aware Transformer with Visual Guidance for Automated 3D Annotation"
_ICLR.cc/2025/Conference — Submitted to ICLR 2025_

### Official Review · Reviewer_zb38 · 2024-10-15

**Soundness:** 2
**Presentation:** 2
**Contribution:** 2
**Rating:** 5
**Confidence:** 4

**Summary:**

This paper introduces MMCAT, a framework designed for automated 3D annotation using multimodal data. By combining point cloud data from LiDAR with images and 2D bounding boxes, MMCAT improves the annotation quality. It integrates specialized encoders for point clouds, images, and 2D boxes, allowing effective feature alignment and multimodal fusion. The model is validated on the KITTI and Waymo Open datasets, achieving SOTA performance in generating 3D annotations, particularly excelling in challenging scenarios.

**Strengths:**

1.  The use of 2D visual cues from images and 2D boxes to guide 3D point cloud annotation effectively addresses the limitations of sparse point cloud data, leading to improvements in accuracy for hard samples.
2. Qualitative results demonstrate the effectiveness of MMCAT.

**Weaknesses:**

1. Although new modalities are introduced as inputs, there aren't many technical contributions to the multimodal transformer itself.
2. 2D bounding boxes are not always available, making MMCAT cannot be applied to 3D raw point clouds.

**Questions:**

What is the effect of inaccurate 2D bounding boxes as inputs, e.g., annotating images with an off-the-shelf 2D detection model?

---

### Official Review · Reviewer_PPoH · 2024-11-03

**Soundness:** 2
**Presentation:** 2
**Contribution:** 2
**Rating:** 5
**Confidence:** 4

**Summary:**

This paper proposes an automatic 3D bounding box annotation method based on a designed multimodal context-aware transformer, termed MMCAT. MMCAT utilizes 2D images and the corresponding 2D bounding boxes as visual cues to guide the regression of 3D bounding boxes. Ultimately, this work employs MMCAT to annotate the training sets of KITTI and Waymo, achieving results on the car/vehicle category that are approximately equivalent to manual annotations.

**Strengths:**

MMCAT has designed a multimodal context-aware transformer consisting of four types of encoders, which makes more comprehensive use of multimodal information. MMCAT uses 2D bounding boxes to optimize the regression of 3D bounding boxes, which can reduce the labeling cost to a certain extent.

**Weaknesses:**

According to the experimental setup of the paper, the authors used manually annotated 2D boxes and some 3D annotation information. In fact, this still represents a non-negligible cost. This paper only provided results for the car/vehicle category using MMCAT, lacking comparative experiments for other categories. Lacks comparison with state-of-the-art automatic annotation algorithms, such as DetZero[1]. The design of the annotator is similar to ViT-WSS3D[2]. However, the annotation cost required is higher, and the contribution of the proposed method is limited. The method proposed in this paper seems to rely on the four modality encoders designed by the MMCAT. Would the use of existing pre-trained encoders affect the performance of MMCAT?
[1] DetZero: Rethinking Offboard 3D Object Detection with Long-term Sequential Point Clouds, ICCV 2023.
[2] A Simple Vision Transformer for Weakly Semi-supervised 3D Object Detection. ICCV 2023

**Questions:**

The author provided the accuracy on the KITTI test set in the paper, but the corresponding accuracy was not found on the KITTI benchmark. What information do images and 2D bounding boxes provide for 3D bounding box regression? How effective is using only a Point+2D encoder? The design of the annotator is similar to ViT-WSS3D; what are the advantages of the proposed method over ViT-WSS3D? The method proposed in this paper seems to rely on the four modality encoders designed by the MMCAT. Would the use of existing pre-trained encoders affect the performance of MMCAT?

---

### Official Review · Reviewer_fuGs · 2024-11-08

**Soundness:** 3
**Presentation:** 3
**Contribution:** 3
**Rating:** 6
**Confidence:** 3

**Summary:**

## Summary
The authors propose a framework that uses pseudo labels obtained from three modalities, LiDAR point clouds, images, and 2D bounding boxes to train 3D object detectors that produce 3D bounding boxes as outputs. Utilizing dense image features in addition to point and 2D box data allows their framework to be robust to challenging cases of heavy occlusion and truncation. They outperform existing weakly-supervised baselines on challenging cases within the KITTI and Waymo datasets.

**Strengths:**

1. *Impact*: Utilizing relatively abundant data modalities for prediction of 3D pseudo labels would enable us to train better 3D object detection models cheaply
2. *Leveraging multi-modal data*: Aligning 3D point cloud and 2D image/bounding box data to obtain more accurate 3D labels is a useful research direction given the abundance of 2D data and maturity of image encoders.

**Weaknesses:**

1. Table 1 shows that their method is unable to beat previous SoTA on easier cases within KITTI. I would expect their approach to perform at least as well as, other weakly supervised methods that use 2D data. Could the authors discuss why their method is better on challenging cases but cannot beat the SoTA on the easier cases in KITTI (Table 1)? An analysis of failure modes to explain this behavior would be helpful for the community.

**Questions:**

1. What is the rationale for a uniform architecture design across modalities in the MMCAR architecture (Section 3.2)? Wouldn't different modalities benefit from modality-specific architectural designs?
2. Why does the third column (Full Supervision) say "2D Box" for MMCAT in Tables 2 and 4? Isn't MMCAT also trained on 3D bounding boxes as supervision during its training phase as described in Section 3.4?

---

### Meta-Review · Area_Chair_7SEP · 2024-12-22

**Metareview:**

The paper proposes MMCAT, a framework for automated 3D bounding box annotation by leveraging multimodal data, including LiDAR point clouds, images, and 2D bounding boxes. The major strength of the paper is that it demonstrates the promise for reducing reliance on manual annotation by leveraging multimodal data, where there are abundant in the real world. On the negative side, the reviewers are concerned about the similarity to previous work, narrowed scope of the method (only focusing on the car/vehicle category), and incomplete benchmarking. The reviewers are on the fence. After thorough discussion, while the reviewers appreciate the use of multimodal data to help the annotation, they are still worried about the experimental evaluation and the key difference to prior art. The ACs agree with the reviewers. The authors are encouraged to incorporate the feedbacks from the reviewers and resubmit to a future venue.

**Additional Comments On Reviewer Discussion:**

The reviewers were primarily concerned about the similarity to prior work and incomplete experimental evaluations. While the authors made quite a bit of attempts, the reviewers are still worried about the lack of experimental analyses (ie, missing the results of several categories) and key differences.

---

### Decision · Program_Chairs · 2025-01-22

Reject